# THE CROSSOVER STRATEGY BASED ON THE CELLULAR AUTOMATA FOR GENETIC ALGORITHMS WITH BINARY CHROMOSOMES POPULATION

## ABSTRACT

In this paper we propose a crossover operator for genetic algorithms with binary chromosomes population based on the cellular automata (*CGACell*). After presenting the fundamental elements regarding cellular automata with specific examples for one- and two- dimensional cases, the the most widely used crossover operators in applications with genetic algorithms are described and the crossover operator based on cellular automata is defined. Specific forms of the crossover operator based on the ECA and 2D CA cases are described and exemplified. The *CGACell* crossover operator is used in the genetic structure to improved the KNN algorithm in terms of the parameter represented by the number of nearest neighbors selected by the data classification method. Validity and practical performance testing is performed on image data classification problems by optimizing the nearest-neighbors-based algorithm. The experimental study on the proposed crossover operator, by comparing the algorithm based on *CGACell* with standard data classification algorithms such as PCA, Kmeans or KNN, attests good qualitative performance in terms of correctness percentages in the recognition of new images, in classification applications of facial image classes corresponding to several persons.

## 1 INTRODUCTION

The numerous theoretical and practical applications, conferred by the ability to offer innovative solutions for a varied range of complex problems, have positioned optimization techniques in the attention of researchers from various fields such as machine learning, operations research, computational systems biology, mechanics or economics and finance. Among the categories of optimization techniques, we can specify stochastic optimization, linear programming, quadratic programming, continuous optimization, discrete optimization, unconstrained optimization, as well as evolutionary algorithms that are able to find solutions in complex and large data spaces (4), (18), (32). The main evolutionary algorithms used are: Genetic Algorithms, Particle Swarm Optimisations (PSO), Ant Colony (AC), Simulated Annealing (SA), Immune Algorithms (IA), Artificial Bee Colony (ABC), Firefly Algorithm (FA) and Differential Evolution (DE). Realizing the percentages of use in applications, with a value of over 50%, genetic algorithms were chosen (48). Among the various applications of genetic algorithms, the most important is the optimization of problems by determining the optimal solutions. Genetic algorithms are optimization algorithms inspired by the process of natural selection through which representative individuals are advantaged and were developed by Goldberg (1989) (18). Applications using GA include: data clustering and mining, neural networks, image processing, feature selection for machine learning, medical science, learning robot behavior, traveling salesman problem, vehicle routing problem, financial markets, manufacturing system or mechanical engineering design (24), (36). In the process of searching for solutions in the solution space, a genetic algorithm performs an evolutionary transformation of the population, over several generations, by using the genetic operators of selection, crossover and mutation, and the quality of the descendants of the chromosomes in the current population is evaluated by the function fitness that decides the composition of the new population of chromosomes. An essential role in the evolutionary process of a genetic algorithm is represented by the reproduction stage in which the chromosomes, selected to participate in the formation of the new generation, determine the new

descendants through the gene crossing operation (24), (38), (50). Thus, the crossover problem has many developments in specialized research, establishing experimental performances from classic methods (with one point, with two or more crossing points, uniform or order) to techniques to solve object classification problems (Looseness control crossover or Greedy partition crossover) (5), (29), (30), (45), (47). The aim of the paper is to propose a crossover operator for genetic algorithms using cellular automata and to establish the experimental performance in specific data classification applications.

## 2 THE CELLULAR AUTOMATA

Cellular automata are among the oldest models of natural computing, the first studies being carried out by John von Neumann in the $1940$s. Cellular automata have a biological motivation rendered by the design of artificial systems that have the property of self-replication, having as inspiration the model from the level of the human brain in which memory and processing units do not operate separately but are able to work together (25). At the S. Ulam's suggestions, J. Neumann developed the system by considering a discrete space consisting of a two-dimensional mesh of machines with finite states. Given the ability to solve problems of high complexity, cellular automata have been used in the fields of natural sciences, mathematics or computer science (51; 42). After A. Burks published J. von Neumann's book in 1966, more scientists became interested in cellular automata. John Conway in 1970 introduced a CA called *Game of Life*. He used a two-dimensional lattice with two possible states for the cells (*dead and alive*) and the transition being made based on the neighboring cells of the dead or alive type (17). A cellular system constitutes a basic framework or cellular space in which the events of the automaton can take place and the simple and precise rules applied will ensure the functioning of the system (6). The cellular space is defined by an $n$-dimensional space together with a neighborhood relation defined on this space. The neighborhood relation associates each cell in the cell space with a lot of neighboring cells. A cellular automata is specified by a finite list of states for each cell, an initial state and a transition function that, based on the neighborhood relationship at time $t$, establishes a new state at time $t + 1$ corresponding to each cell. In the standard formulation, CA can be studied in the $\mathbb{Z}^d$ ($d \in \mathbb{N}^*$) space and using an alphabet $L$. In recent years, several generalizations relative to the CA alphabet have been developed through the use of groups, vector spaces or commutative monoids (8), (9), (10), (53). For the case of a group $V$ and an alphabet set $L$, the specific elements (configuration space, shift action, memory set and local function) for defining a cellular automaton are given below (7), (26). The configuration space represents the set of functions defined on $V$ with values in $L$ of the form $L^V = \{f : V \to L\}$.

The shift action of $V$ on $L^V$ is defined by $g \cdot f(h) = f \cdot (g^{-1}h)$, for all $f \in L^V$, $g, h \in V$.

The memory set represents a subset $M$ of the set $V$ ($M \subseteq V$) and a local function is defined by a function $\varphi : L^M \to L$.

A cellular automaton is defined by a function $\gamma : L^V \to L^V$ that satisfies the property $\gamma(f)(g) = \varphi(g^{-1} \cdot f)|_M)$, with memory set $M$ and local function $\varphi$, $\forall f \in L^V$, $g \in V$ and where $|_M$ denotes the restriction to $M$ of a configuration in $L^V$.

**Elementary Cellular Automaton (ECA) 1D CA** The elementary cellular automaton represents a one-dimensional cellular automaton (1D CA) with two states ($L = \{0, 1\}$), and the transition rule in a new state of a cell depends only on the neighborhood formed with the adjacent cells (the current cell and the left and right neighbors of the cell). An ECA capable of universal computation due to the property of simulating any Turing machine, and thus the system is able to recognize or decide other sets of data manipulation rules, is a one-dimensional CA corresponding to rule 110 of the Wolfram Code, establishing the resulting states for the eight possible configurations associated for a cell with its two adjacent neighbors. To establish the particular case of the ECA, the settings corresponding to the one-dimensional case with two cell states are considered by: $V = \mathbb{Z}$ and $L = \{0, 1\}$. With these conditions, the configuration space $L^V = L^{\mathbb{Z}} = \{f : \mathbb{Z} \to L\}$, $L^{\mathbb{Z}} = \{\dots, f^{-n}, f^{-(n-1)}, \dots, f^{-2}, f^{-1}, f^0, f^1, f^2, \dots, f^n, \dots\}$, where $f^i = \{\dots, f_{-m}, f_{-(m-1)}, \dots, f_{-2}, f_{-1}, f_0, f_1, f_2, \dots, f_{m-1}, f_m, \dots\}$, $f_i = f(i)$, $\forall i \in \mathbb{Z}$ and $f(i)$ represents the value of the image of $i$ by the function $f^i$, $n, m \in \mathbb{N}$. In this case the shift action of $r \in \mathbb{Z}$ on $L^{\mathbb{Z}}$ can be written by the following relation $r \cdot f = \{\dots, f_{-r-m}, f_{-(r+m-1)}, \dots, f_{-r-2}, f_{-r-1}, f_{-r}, f_{-r+1}, f_{-r+2}, \dots, f_{-r+m-1}, f_{-r+m}, \dots\}$. Also, the memory set $M$ is established based on the pattern

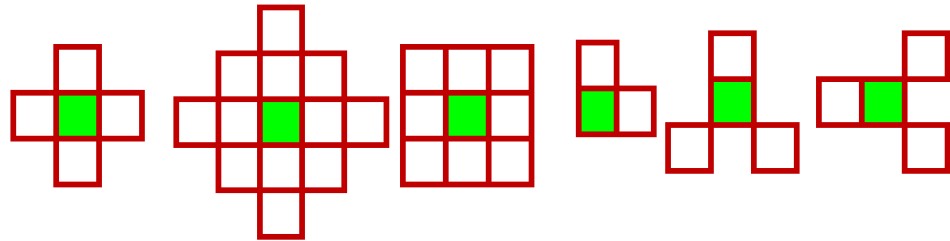

Figure 1: Neighborhood templates used for 2D CA - in order two von Neumann neighborhoods with 5 neighbors and extended to 13 neighbors, Moore template with 9 neighbors, Smith and Cole (two images) neighborhoods.

associated with the neighborhood used for the cells of the automaton by $M = \{-1, 0, 1\}$ and the local function of the automaton is defined by $\varphi : L^M \to L$ ($\varphi : \{0,1\}^{\{-1,0,1\}} \to \{0,1\}$).

The cellular automaton is defined based on set memory $M$ and the local function $\varphi$ through the function $\gamma : \{0,1\}^{\mathbb{Z}} \to \{0,1\}^{\mathbb{Z}}$ that satisfies the property $\gamma(f)(j) = \varphi(j^{-1} \cdot f)|_M)$, $\forall f \in \{0,1\}^{\mathbb{Z}}$, $j \in \mathbb{Z}$ and where $|_M$ denotes the restriction to $M = \{-1, 0, 1\}$ of a configuration in $\{0,1\}^{\mathbb{Z}}$.

**Two-dimensional cellular automata (2D CA)** There are several approaches in the field to define two-dimensional cellular automata (2D CA). As in the case of ECA, the components of a cellular automaton are the lattice (set of cells - each one in a state), the neighborhood and the local transition rules. The rules represent the communication between each cell and its neighborhood, it is local, uniform for the whole lattice and synchronous. The rules determine the global evolution of the system at each discrete step in time. The lattice represents the $V$ group also considered for the definition of one-dimensional cellular automata ECA. If $V = \mathbb{Z}$ was considered for ECA, $V = \mathbb{Z}^2$ is considered for the definition of two-dimensional cellular automata. The neighborhood is the set of cells taken into account in the evolution of the cell. The most used are the Von Neumann and Moore neighborhoods (34). The von Neuman neighborhood model is a template that contains the immediate north, south, east, and west neighbors of the current cell. The Moore neighborhood model is a template that includes all the immediate neighbors of the current cell (from the north, south, east, west, northeast, northwest, southeast and southwest). Over time, several models with adequate results in applications have been added to these templates (49), (13) (Figure 1). The states of a 2D CA at any moment can be represented by a binary matrix $A$ of size $m \cdot n$, $m, n \in \mathbb{N}^*$ that will contain the cells of the automaton. According to the definition in (34), for the von Neumann neighborhood consisting of five neighbors and the lattice $A$, the state of a cell in position $(i, j)$ $(i, j \in \mathbb{N}^*)$ of $A$, $a_{ij}^{t+1}$ is updated from a new generation $t + 1$ ($t \in \mathbb{N}^*$) through a relation of the form: $a_{ij}^{t+1} = \phi(a_{ij-1}^t, a_{ij+1}^t, a_{ij}^t, a_{i-1j}^t, a_{i+1j}^t)$ with $i = \overline{1..n}$, $j = \overline{1..m}$, $t = \overline{1..N_g}$, where $N_g \in \mathbb{N}^*$ is the number of generations of evolution of the automaton. Given the size of the considered neighborhood with $k = 5$ neighbors, we have $2^k$ possible states associated with cells in the neighborhood based on a rule for associating bits from the binary representation of numerical values in the interval $[0, 2^{2^k} - 1] \cap \mathbb{N}$.

Another variant used is the one in which the state of the current cell is determined by a rule based on a function $f$ with the argument represented by the sum of the values of the states of the neighbors retrieved according to the template used $a_{ij}^{t+1} = f(a_{ij-1}^t + a_{ij+1}^t + a_{ij}^t + a_{i-1j}^t + a_{i+1j}^t)$ with $i = \overline{1..n}$, $j = \overline{1..m}$ and $t = \overline{1..N_g}$ with the extension of $\tilde{f}$ through $a_{ij}^{t+1} = \tilde{f}(a_{ij}^t, a_{ij-1}^t + a_{ij+1}^t + a_{i-1j}^t + a_{i+1j}^t)$ by using only the four neighbors. In this case the sum of the binary state of the neighbors are maximally 4 and we may establish that the new state of the cell $a_{ij}^{t+1}$ to be equal with the bit value by position $a_{ij-1}^t + a_{ij+1}^t + a_{i-1j}^t + a_{i+1j}^t$ from the binary representation for the choose rule in numerical domain like $[0, 2^k - 1] = [0, 2^5 - 1] \cap \mathbb{N}$.

## 2.1 Crossover methods used by GA

Crossover operators are used to generate offspring chromosomes by recombining selected individuals from the current population to form the new generation. Among the most used crossover oper-

ators are single point, two-point, k-point, uniform, partially matched, order, precedence preserving crossover, shuffle, reduced surrogate and cycle (24), (29).

# 3 CGACELL OPERATOR FOR BINARY CHROMOSOMES POPULATION OF GENETIC ALGORITHMS

In this section, the crossover operator based on cellular automata is introduced (*CGACell operator*) for the case of chromosomes represented by genes with binary values.

Consider a cellular automaton $\gamma$ and a genetic algorithm $G$ with the population consisting of genes with binary values $P(t) = \{ch_1, ch_2, \ldots, ch_{nc}\}$, where $ch_i$ is the $i$ chromosome, $ch_i = \{g_{i1}, g_{i2}, \ldots, g_{ng}\}$, $g_{iq} \in \{0, 1\}$ for $q = \overline{1..ng}$ and $i = \overline{1..nc}$, $nc \in \mathbb{N}^*$ is the number of chromosomes in $P(t)$ at time $t \in \mathbb{N}^*$ and $ng \in \mathbb{N}^*$ is the number of genes of the chromosomes $ch_i$ with $i \in \mathbb{N}^*$.

[**CGACell Crossover operator**] The crossover operator *CGACell* is defined for a population of binary chromosomes $P(t) = \{ch_1, ch_2, \ldots, ch_{nc}\}$, a cellular automaton $\gamma$ ($k$- dimensional, $k \in \mathbb{N}^*$) and a neighborhood $V_h$ established according to a given template for taking neighbors, $V_h = \{v^1, v^2, \ldots, v^{n_r}\}$, $v^i \in \mathbb{N}^k$, $i = \overline{1..n_r}$, $nr \in \mathbb{N}^*$ is the elements number of $V_h$, as follows:

$$CGACell(C_s, V_h, \gamma) = kD\ CA(C_s, V_h, \gamma), \tag{1}$$

where $kD\ CA(C_s, V_h, \gamma)$ represents the descendant chromosome obtained by applying a $k$-dimensional cellular automaton ($kD\ CA$) on a set of chromosomes $C_s$ (with $C_s \subset P(t)$) chosen by the selection techniques established by GA, with a cardinality correlated with the size of the neighborhood used $V_h$.

[**CGACell ECA**] In the one-dimensional case of cellular automata, the crossover operator *CGACell ECA* is obtained. We have the relationship $CGACell(C_s, V_h, \gamma) = 1D\ CA(C_s, V_h, \gamma)$, where the neighborhood $V_h$ consists of 3 elements represented by the neighboring cells on the left and right of the current cells, $V_h = \{v^1, v^2, v^3\}$, $v^i \in \{0, 1\}$, $n_r = 3$, $i = \overline{1..n_r}$, $nr \in \mathbb{N}^*$ is the elements number of $V_h$.

The *CGACell ECA* operator can be applied in two ways: individually on each chromosome chosen in the selection stage for recombination, or mixed on genes on the same positions (or corresponding through a mapping) from a number of chromosomes chosen for recombination equal to the cardinality of the set of memory $M$ of the cellular automaton (equal to three for the exemplified case).

**Example 1 - CGACell ECA** The method of applying the operator *CGACell ECA* at the chromosome level using an ECA with $rule\ 110$ previously presented in section CA ($remark\ 1$) is illustrated in Figure 2. Also, the results of the *CGACell ECA* crossover operator for $ECA\ Rule\ 90$ are illustrated in Figure 3. The two offspring chromosomes correspond to the chromosomes chosen by selection to participate in reproduction for GA. Cells that change their state by applying *CGACell ECA* Crossover and according to $ECA\ Rule\ 110$ or $90$ are highlighted on a white colored background for each descendant. For cells in extreme positions, they are generically filled with inactive values equal to zero for full ECA rule application.

| CGACell Crossover operator - ECA Rule 110 | | | | | | | | | Dec. val. |
|---|---|---|---|---|---|---|---|---|---|
| **Parent 1:** 1 | 0 | 0 | 1 | 0 | 0 | 1 | 0 | 1 | 293 |
| **Parent 2:** 0 | 1 | 1 | 0 | 0 | 1 | 0 | 0 | 1 | 201 |
| ↓↓ D1= CGACell ECA(Parent 1) ↓↓ D2= CGACell ECA(Parent 2) ↓↓ | | | | | | | | | |
| **Descend. D1:** 1 | 0 | 1 | 1 | 0 | 1 | 1 | 1 | 1 | 367 |
| **Descend. D2:** 1 | 1 | 1 | 0 | 1 | 1 | 0 | 1 | 1 | 475 |

Figure 2: The *CGACell ECA* Crossover for two parents with 9 genes by applying ECA Rule 110 at the individual chromosome level

**Example 2 - CGACell ECA** The crossover operator *CGACell ECA* can also be applied in mixed mode on three chromosomes, chosen after the selection stage. Let the set of chromosomes chosen

| CGACell Crossover operator - ECA Rule 90 | | | | | | | | | Dec. val. |
|---|---|---|---|---|---|---|---|---|---|
| Parent 1: | 1 | 0 | 0 | 1 | 0 | 0 | 1 | 0 | 1 | 293 |
| Parent 2: | 0 | 1 | 1 | 0 | 0 | 1 | 0 | 0 | 1 | 201 |
| | ↓↓ *D1= CGACell ECA(Parent 1)* ↓↓ *D2= CGACell ECA(Parent 2)* ↓↓ | | | | | | | | | |
| Descend. D1: | 0 | 1 | 1 | 0 | 1 | 1 | 0 | 0 | 0 | 216 |
| Descend. D2: | 1 | 1 | 1 | 1 | 1 | 0 | 1 | 1 | 0 | 502 |

Figure 3: The *CGACell ECA* Crossover for two parents with 9 genes by applying ECA Rule 90 at the individual chromosome level

for reproduction in the GA selection stage be of the form $C_s = \{ch_1, ch_2, ch_3\}$ from the current population $P(t)$.

Having the representation of chromosomes $ch_i = \{g_{i1}, g_{i2}, \ldots, g_{ing}\}$, $g_{iq} \in \{0, 1\}$ for $q = \overline{1..ng}$ ($ng = 9$) and $i = \overline{1..nc}$, $nc = 3$, *the neighborhood* $V_h$ is formed for the application of the *CGACell ECA* crossover operator, by $V_h = \{v^1, v^2, v^3\}$, with $v^i = g_{ij}$, $j \in \{1, 2, \ldots, ng\}$, $v^i \in \{0, 1\}$, $n_r = 3$, $i = \overline{1..n_c}$.

The application of the operator *CGACell ECA* at a mixed level on three chromosomes using ECA with $rule$ 110 is presented in Figure 4 and by using ECA with $rule$ 90 in Figure 5. We can see the descendant chromosome ($D1$ or $D2$) obtained by *CGACell ECA* crossover using $ECA\ Rule$ 110 or 90 with the neighborhood formed by taking the values of the genes on the same positions from the three chromosomes selected for crossing.

Descendant chromosome ($D1$ or $D2$) cells that differ from the corresponding states in chromosome two, after applying *CGACell ECA* Crossover and $ECA\ Rule$ 110 or 90, are highlighted on a white background. [**CGACell 2D CA**] By using two-dimensional cellular automata,

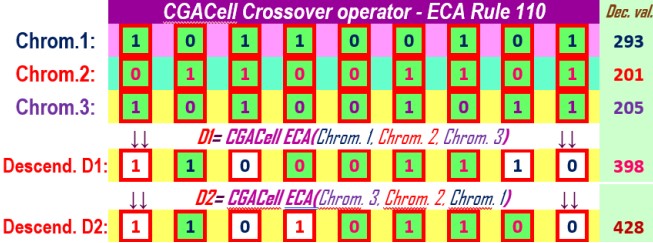

| CGACell Crossover operator - ECA Rule 110 | | | | | | | | | Dec. val. |
|---|---|---|---|---|---|---|---|---|---|
| Chrom.1: | 1 | 0 | 1 | 1 | 0 | 0 | 1 | 0 | 1 | 293 |
| Chrom.2: | 0 | 1 | 1 | 0 | 0 | 1 | 1 | 0 | 1 | 201 |
| Chrom.3: | 1 | 0 | 1 | 0 | 0 | 1 | 0 | 1 | 1 | 205 |
| | ↓↓ *D1= CGACell ECA(Chrom. 1, Chrom. 2, Chrom. 3)* ↓↓ | | | | | | | | | |
| Descend. D1: | 1 | 1 | 0 | 0 | 0 | 1 | 1 | 1 | 0 | 398 |
| | ↓↓ *D2= CGACell ECA(Chrom. 3, Chrom. 2, Chrom. 1)* ↓↓ | | | | | | | | | |
| Descend. D2: | 1 | 1 | 0 | 1 | 0 | 1 | 1 | 0 | 0 | 428 |

Figure 4: The *CGACell ECA* Crossover in the mixed variant for three chromosomes with 9 genes by applying ECA Rule 110

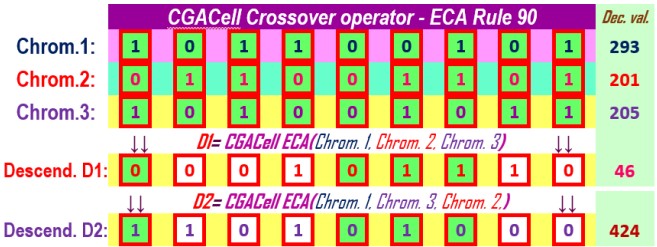

| CGACell Crossover operator - ECA Rule 90 | | | | | | | | | Dec. val. |
|---|---|---|---|---|---|---|---|---|---|
| Chrom.1: | 1 | 0 | 1 | 1 | 0 | 0 | 1 | 0 | 1 | 293 |
| Chrom.2: | 0 | 1 | 1 | 0 | 0 | 1 | 1 | 0 | 1 | 201 |
| Chrom.3: | 1 | 0 | 1 | 0 | 0 | 1 | 0 | 1 | 1 | 205 |
| | ↓↓ *D1= CGACell ECA(Chrom. 1, Chrom. 2, Chrom. 3)* ↓↓ | | | | | | | | | |
| Descend. D1: | 0 | 0 | 0 | 1 | 0 | 1 | 1 | 1 | 0 | 46 |
| | ↓↓ *D2= CGACell ECA(Chrom. 1, Chrom. 3, Chrom. 2.)* ↓↓ | | | | | | | | | |
| Descend. D2: | 1 | 1 | 0 | 1 | 0 | 1 | 0 | 0 | 0 | 424 |

Figure 5: The *CGACell ECA* Crossover in the mixed variant for three chromosomes with 9 genes by applying ECA Rule 90

the version of the *CGACell 2D CA* crossover operator is obtained. We have the relationship $CGACell(C_s, V_h, \gamma) = 2D\ CA(C_s, V_h, \gamma)$, where the neighborhood $Vsh$ is built in accordance with the templates presented for two-dimensional cellular automata (remark 2, figures 1), the most representative being the von Neumann neighborhood with five neighbors, respectively, the Moore

neighborhood with nine neighbors, $V_h = \{v^1, v^2, v^{nv}\}$, $v^i \in \{0, 1\}$, $i = \overline{1..n_r}$ and $nv \in \mathbb{N}^*$ is the number of neighbors in $V_h$.

The configuration space of the cellular automaton $V$ is formed by completing the two-dimensional space with chromosomes chosen $(C_s)$ from the current population $P(t)$ to participate in the creation of descendants and the new generation of individuals.

**Application in data classification with KNN method of the *CGACell* crossover-based genetic algorithm** It is considered a data classification application using the KNN algorithm based on the nearest neighbors (2), (3), (21). Performance testing in data classification through the KNN algorithm is performed on the basis of a set of training data $X^L$ and a set of test data $X^t$, both sets of data from multiple data classes, and the key element that determines the improvement of the results is the parameter $k$ number of neighbors taken into account for classification. For the optimal establishment regarding the performance in classification of the value of the $k$ parameter, it can be achieved by applying a genetic algorithm based on *CGACell* crossover. The genetic algorithm based on the *CGACell* crossover operator (usually the ECA or 2D CA versions are used), denoted by *CGACell-GA* for data classification by the nearest neighbors method includes the following work steps:

+ The population is made up of chromosomes with binary values corresponding to the binary representation of the values in the field of representation of the $k$ parameter that designates the number of nearest neighbors that will decide, depending on the classes of origin, the classification results for the KNN algorithm.

+ The population consists of binary chromosomes with a size equal to the number of bits $(nb \in \mathbb{N}^*)$ in the representation of the maximum value $(K_{max})$ in the range of possible values for the parameter $k$. Let $K_{max}$ be the maximum value of $k$ established based on the number of data used for training by $K_{max} = pk * \sum_{i=1}^{l} n_i^L$, $(k \in [K_{min}, K_{max}])$, where $n_i^L$ is the number of the training data input from class $i$, $i = \overline{1..l}$, $l \in \mathbb{N}^*$ is number of data classes, and $pk$ is the selection weight for the maximum number of neighbors with values, usually chosen, in the interval $pk \in [0.4, 0.8]$ and $K_{min} = 0$.

+ The population of the genetic algorithm consists of chromosomes as follows: $P(t) = \{ch_1, ch_2, ..., ch_{nc}\}$, where $ch_i$ is the $i$ chromosome, $ch_i \in [K_{min}, K_{max}]$, $ch_i = \{b_1, b_2, ..., b_{nb}\}$, $b_q \in \{0, 1\}$ for $q = \overline{1..nb}$ and $i = \overline{1..nc}$, with $nc \in \mathbb{N}^*$ is the number of chromosomes and in the experiments a value adapted to the total number of training data was used.

+ The fitness function $f(ch_i)$, $i = \overline{1..nc}$ is represented by the performance (percentage of correct classification) in the classification of the test data obtained by using a number of nearest neighbors equal to the value in base ten of the chromosome $(ch_i)$ argument of the function.

+ The selection is carried out by the roulette type method after determining the scaling function for the chromosomes in the current population (the moment of time $t \in \mathbb{N}^*$), establishing the selection probabilities and the actual selection of chromosomes $P_1(t) = \{ch_{i_1}, ch_{i_2}, ..., ch_{i_{nsc}}\}$ with $\{i_1, i_2, ..., i_{nsc}\} \subset \{1, 2, ..., nc\}$ and $nsc \in \mathbb{N}^*$ is the number of selected chromosomes for the crossing stage based on randomly generated values in the numerical range $[0, 1]$.

+ The *CGACell* crossover is performed for the chromosomes selected from the set $P_1(t)$ through several transformation methods at the level of the binary vectors from the chromosome representations. *CGACell* ECA or 2D CA crossover are used. For each crossing case, the corresponding experimental results were established in the classification of the test data from the test set $X^t$.

+ The mutation is carried out at the level of the chromosomes in the set $P_1(t)$ resulting after the step of crossing the binary genes. The mutation operation involves updating certain genes, in a very small proportion (between $1\% - 5\%$) by transforming the chosen genes into the complementary binary value.

+ After the genetic mutation operation, the set of chromosomes in $P_1(t) \cup P(t)$ is reevaluated by applying the fitness function in order to establish their quality and the new generation of chromosomes is formed by choosing the best $nc$ chromosomes.

+ The algorithm is repeated by applying the genetic operators of selection, *CGACell* crossover and mutation and forming new generations with the best performing $nc$ chromosomes until a predetermined maximum number of training generations is reached or the optimal value is reached or in the situation where the classification performance test data stagnates.

The experimental results regarding the application of *CGACell* crossover in KNN data classification showed good performance in a study compared to other classification methods and are presented in the experimental analysis section.

## 4 RELATED WORK

In the specialized literature there are many optimization applications in which genetic algorithms are used with improvements at the crossover stage. E. Tafehi et al. introduced a new and improved method for GA based on Chaotic Cellular automata (CCA) along with influencing pseudo random number generator. The operations specific to the genetic algorithm of selection, mutation and crossover are influenced by pseudo random number generator and thus change the behavior of the evolution of chromosomes in the exploration space (40). U. Cerruti et al. introduced an original implementation of a cellular automaton whose rules use a fitness function to select for each cell the best mate to reproduce and a crossover operator to determine the resulting offspring. They also created two generalizations of the game of life and other applications in the paper (11). Deng et al. in the paper (14) proposed a hybrid cellular genetic algorithm with simulated annealing in which the genetic algorithm used together with cellular automata. They applied a hybrid cellular genetic algorithm combined with a simulated annealing algorithm to solve the TSP, and the experimental results showed the optimization performance of the hybrid cellular genetic algorithm. In the paper (28) M. Mitchell et al. have applied genetic algorithms to the design of cellular automata that can perform calculations that require global coordination. The proposed work method establishes for each chromosome in the population a candidate rule table. The evolution of CA with GA has provided an appropriate framework in which to study the mechanisms by which an evolutionary process could create complex coordinated behavior in decentralized distributed natural systems. S.J. Louis and G.L. Raines used a genetic algorithm to calibrate a cellular automaton that modeled mining activity on public land in Idaho and Western Montana. The genetic algorithm searches in the parameter space of the transition rules of a 2D CA to find the parameters of the rule that matches the observed mining activity data (27).

## 5 EXPERIMENTAL ANALYSIS FOR *CGACell* CROSSOVER OPERATOR IN SPECIFIC TASKS

The performance testing of the *CGACell* crossover operator based on cellular automata is performed in two-dimensional data classification applications (images). The *CGACell* crossover operator presented in this paper is used in image classification problems through the KNN algorithm to optimize the parameter $k$ representing the number of nearest neighbors. The genetic algorithm (*CGACell-GA*) based on *CGACell* crossover is detailed in remark 5. The Yale database contains 165 images

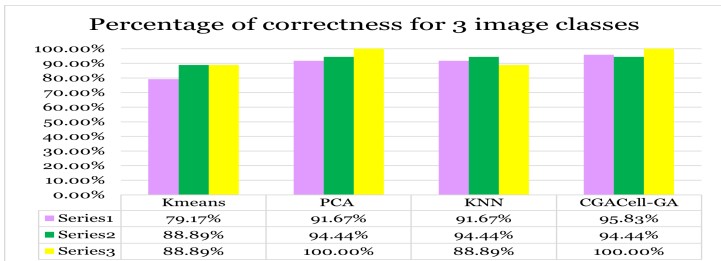

Figure 6: Case I - Image classification for three face image classes from the Yale face database

of faces corresponding to 15 people with 11 images each representing certain facial expressions or

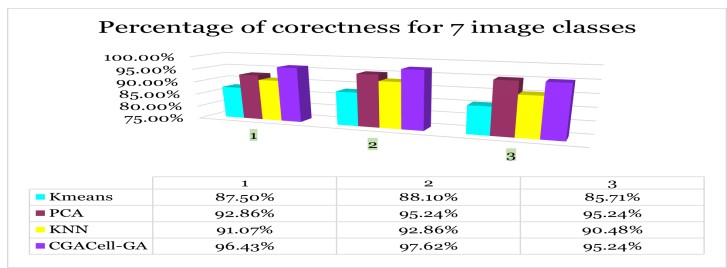

| | 1 | 2 | 3 |
|---|---|---|---|
| Kmeans | 87.50% | 88.10% | 85.71% |
| PCA | 92.86% | 95.24% | 95.24% |
| KNN | 91.07% | 92.86% | 90.48% |
| CGACell-GA | 96.43% | 97.62% | 95.24% |

Figure 7: Case II - Image classification for seven face image classes from the Yale face database

configurations (*glasses, happy, left light, right light, sad, sleepy, surprised, wink, center light, no glasses* and *normal*). The experimental results are achieved by the stability of the correctness percentages in the classification of image classes from the Yale image database. The performances in testing new images for the proposed algorithm are highlighted through an experimental comparative study with standard classification methods such as the KMeans algorithm, the Principal Component Analysis (PCA) algorithm and the standard KNN algorithm. In the comparative study, three cases are considered regarding the number of classes of images: the classification of 3 classes of images, the classification of 7 classes of images and, respectively, all the classes (11) of images available in the database. For each case of the number of classes of images, the correctness percentages in the classification of images corresponding to the use of three, five, and eight training images from each class of images are established. In all three cases, the number of images used for the training stage is represented by the labels *Series1*- three training images, *Series2*- five training images, and *Series3*- eight training images, in figures 6 - 9. Figure 9 shows results obtained from the experimental anal-

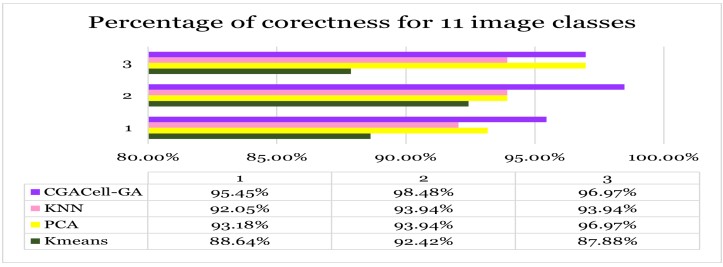

| | 1 | 2 | 3 |
|---|---|---|---|
| CGACell-GA | 95.45% | 98.48% | 96.97% |
| KNN | 92.05% | 93.94% | 93.94% |
| PCA | 93.18% | 93.94% | 96.97% |
| Kmeans | 88.64% | 92.42% | 87.88% |

Figure 8: Case III - Image classification for eleven face image classes from the Yale face database

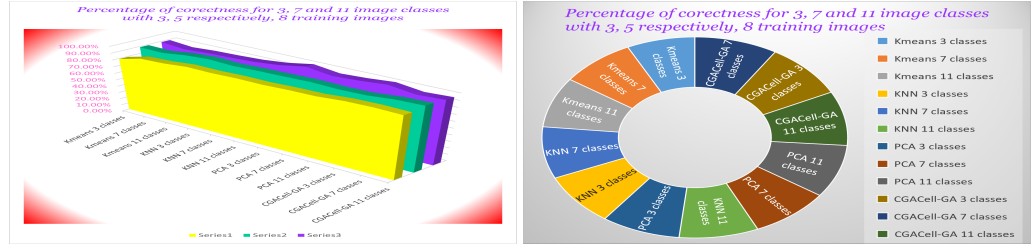

Figure 9: Graphic representation of correct classification percents of face images for three, seven, and eleven classes of face images, respectively, and having three, five, and eight training images, respectively

ysis based on the Yale image database for the classification of images from 3, 7 and 11 face image classes by using a name of 3, 5, respectively 8 training face images. Also, the numerical results obtained from the experimental analysis for the algorithm based on *CGACell* crossover, using the Yale image database for classifying images from face image classes 3, 7, and 11 and having successive

values of 3, 5, and 8 training face images, respectively, and the rest of the images remained in the test set, are specified in table 1.

Table 1: Values of correct classification percents of face images from the Yale database for three, seven, and eleven classes of face images, and having three, five, and eight training images

| Algorithm classification | No. of img. classes | No. of training images | | |
|---|---|---|---|---|
| | | $3\,img.tr.$ | $5\,img.tr.$ | $8\,img.tr.$ |
| Kmeans | 3 (27.27%) | 79.17% | 88.89% | 88.89% |
| Kmeans | 7 (46.66%) | 87.50% | 88.10% | 85.71% |
| Kmeans | 11 (73.33%) | 88.64% | 92.42% | 87.88% |
| KNN | 3 (27.27%) | 91.67% | 94.44% | 88.89% |
| KNN | 7 (46.66%) | 91.07% | 92.86% | 90.48% |
| KNN | 11 (73.33%) | 92.05% | 93.94% | 93.94% |
| PCA | 3 (27.27%) | 91.67% | 94.44% | 100.00% |
| PCA | 7 (46.66%) | 92.86% | 95.24% | 95.24% |
| PCA | 11 (73.33%) | 93.18% | 93.94% | 96.97% |
| CGACell-GA | 3 (27.27%) | 95.83% | 94.44% | 100.00% |
| CGACell-GA | 7 (46.66%) | 96.43% | 97.62% | 95.24% |
| CGACell-GA | 11 (73.33%) | 95.45% | 98.48% | 96.97% |

Case I. For the first test, the images from three classes of images taken from the Yale face database are classified by applying the proposed *CGACell-GA* algorithm, as well as the KMeans, standard KNN and PCA algorithms. The percentages of correct classification of images from the three classes are graphically represented in figure 6 for different values (three, five, and eight images, respectively) of the number of training images from each class of face images.

Case II. For the second test, the procedure is similar to the working method of case 1, except that seven classes of images are classified and the results are in figure 7.

Case III. For the third test, the procedure is similar to the working method of cases 1 and 2, except that eleven classes of images are classified and the results are in figure 8.

## 6 SUMMARY AND CONCLUSIONS

In the elaborated work, a version of crossover specific to genetic algorithms by using cellular automata, was described and established. The introduced crossover operator is based on the functionality of cellular automata, and the specification of the operator for the cases most used in applications, of elementary automata, respectively, two-dimensional and with von Neumann and Moore neighborhood, are described. The results of the comparative study are performed for the proposed algorithm based on the cellular crossover operator with other standard image classification methods from the Yale face database and established good performance in the classification of new images from the considered image classes. In the future development, it will be analyzed the possibility of using the proposed crossover operator in an integrated system of multilayer neural networks with applications in image recognition, as well as in applications for generating knowledge assessment tests of different difficulty categories in the educational system.

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
