# OpenReview forum: "The crossover strategy based on the cellular automata for genetic Algorithms with binary chromosomes population"
_ICLR.cc/2024/Conference — Submitted to ICLR 2024_

### Official Review · Reviewer_oJwJ · 2023-10-31

**Soundness:** 2 fair
**Presentation:** 2 fair
**Contribution:** 2 fair
**Rating:** 3
**Confidence:** 4

**Summary:**

The study introduces a novel crossover operator for genetic algorithms that is inspired by cellular automata. It outlines the operator's functionality and specifies its application for elementary automata, including two-dimensional automata with von Neumann and Moore neighborhoods. Comparative analyses were conducted, evaluating the proposed algorithm using the cellular crossover operator against standard image classification methods from the Yale face database. The research demonstrates the algorithm's strong performance in classifying new images within the considered image classes. Additionally, the paper proposes future investigations, including exploring the potential integration of the crossover operator into multilayer neural network systems for image recognition. The study also suggests examining the operator's use in generating knowledge assessment tests of varying difficulty levels in the educational sector.

**Strengths:**

This paper proposed a novel crossover operator for genetic algorithms that is inspired by cellular automata. The experimental results demonstrate the algorithm's strong performance, particularly in achieving high correctness percentages in recognizing new images for facial image classification tasks related to multiple individuals.

**Weaknesses:**

1. Insufficient presentation: Some concepts are hard to comprehend without explaining clearly.
2. Experiment limitation: Comparative experiments and ablation study are not enough, resulting in there is a gap between results of conclusion.

**Questions:**

1. There are many traditional machine learning methods for handling image classification tasks, i.e., SVM, K-tree, MLP, etc. The author only compares the proposed method with KNN, PCA, and KMeans. The comparative experiments are not enough to verify its performance.
2. The experiment only shows the difference between CGACell-GA and its competitors. It does not analyze the advantage of cellular automata strategy compared to other crossover operators. This research lacks the ablation study to discuss the impact of automata, which leaves a gap between results and conclusions.
3. The author just tests the performance of all methods on one database. It is a small database including 165 images. It is inadequate to use this database to verify the performance.
4. Although the author gives some examples of rules 110 and 90, the detailed process still is not clear. It is not clear how to work in high-dimensional condition. Some concepts are hard to comprehend without explaining clearly.
5. In experiments, How do the authors divide the database? The partition of the database can affect the results. The authors, therefore, should describe the partition details.
6. The authors should give the details of the compared methods for fair comparison.
7. Please ensure that acronyms and concepts are defined before using them.
8. A massive number of symbols are introduced in the formation. The equation and their variables should be explained in a clearer fashion.

---

### Official Review · Reviewer_3qA6 · 2023-10-31

**Soundness:** 2 fair
**Presentation:** 1 poor
**Contribution:** 1 poor
**Rating:** 1
**Confidence:** 4

**Summary:**

Authors propose a crossover strategy based on cellular automata (CA) behaviour for the simplest binary representation Genetic Algorithm (GA). Rather than performing a single-point or cycle crossover, offspring chromosomes are obtained by arranging parents chromosomes in a grid, and applying CA rules to generate corresponding bits in the descendants' bit strings. Authors provide a brief experimental assessment of their approach by optimizing the number neighbours used in a KNN for a classification task.

**Strengths:**

The idea advanced by authors is somewhat intriguing; however, it remains to be seen if it has some practical applications or advantages over other more well known and conventional crossover methods. Unfortunately, the paper falls very short in providing any evidence on the practical impact, or theoretical implications, of the proposed method.

**Weaknesses:**

- The paper lacks the appropriate tone for a conference article. It feels more as a textbook chapter, or a blog entry aimed at general audiences. It introduces in great details concepts related to CAs, it also spends several lines rambling in general about optimization methods.
- The related work section is missing the main comparison against other approaches. Although authors do mentions several recent works that combine concepts of CAs with GAs, they never contrast their method with those previous works. They never specify what their contribution exactly is, how their proposed approach fit within the context of the related works.
- Authors never introduce the motivation or insight behind their entire work. Although the proposed concept is somewhat interesting, they never really explain what motivated them to attempt such approach, or the gains they expected to obtain. Even if their research was motivated by pure curiosity alone, they should have bring in up, clarify it in some way or another.
- Experimental assessments lacks comparison with proper baselines. Authors perform some experiments in which they utilize their approach to optimize the number of neighbours used in a KNN, and compare against some other classification algorithms and a defaulted KNN. However, since they are proposing a new crossover for binary representation, they should've compared against single point or multi-point crossovers, or even against a hill climbing mechanism.
- The experimental design and results are unintelligible. Authors claim to test their approach in optimizing the parameters for a classification algorithm (the K in a KNN), and perform some experiments in a classification task; but then somehow the competing approaches they present are clustering and dimensionality reduction methods (Kmeans and PCA), which doesn't make any sense.

**Questions:**

I consider this draft way below the quality required for ICLR. As I already mentioned, the manuscript feels more aimed at a textbook or divulgation note. Even if all its weaknesses were to be addressed, it still falls a bit outside the scope of ICLR, since it mostly deals with intricacies related more to Evolutionary Computation.

---

### Meta-Review · Area_Chair_Le74 · 2023-11-27

**Metareview:**

The paper proposes a novel crossover operator for genetic algorithms based on cellular automata rules. It aims to generate offspring chromosomes by arranging parent chromosomes in a grid and applying CA rules. The idea is intriguing but the actual contributions and benefits compared to standard crossover methods are unclear. The motivation and related work comparisons are lacking. Reviewers point out insufficient experimentation, lack of ablation studies, small datasets, and lack of comparisons to other methods as weaknesses. Overall the reviews indicate the idea has potential but the paper falls short on soundness, presentation quality, benchmarking, and novelty of contributions. Major revisions would be needed to make the paper acceptable.

**Justification For Why Not Higher Score:**

There are major concerns, but the authors did not respond.

**Justification For Why Not Lower Score:**

N/A

---

### Decision · Program_Chairs · 2024-01-16

Reject